# TCA cycle remodeling drives proinflammatory signaling in humans with pulmonary tuberculosis

Jeffrey M. Collins[1]*, Dean P. Jones[1], Ashish Sharma[2], Manoj Khadka[3], Ken H. Liu[1], Russell R. Kempker[1], Brendan Prideaux[4], Kristal Maner-Smith[3], Nestani Tukvadze[5], N. Sarita Shah[1,6,7], James C. M. Brust[8], Rafick-Pierre Sékaly[2], Neel R. Gandhi[1,6,7], Henry M. Blumberg[1,6,7], Eric A. Ortlund[3‡], Thomas R. Ziegler[1,9‡]

1 Department of Medicine, Emory University School of Medicine, Atlanta, Georgia, United States of America, 2 Department of Pathology, Emory University School of Medicine, Atlanta, Georgia, United States of America, 3 Department of Biochemistry, Emory University, Atlanta, Georgia, United States of America, 4 Department of Neuroscience, Cell Biology, & Anatomy, University of Texas Medical Branch, Galveston, Texas, United States of America, 5 National Center for Tuberculosis and Lung Diseases, Tbilisi, Georgia, 6 Department of Global Health, Emory University Rollins School of Public Health, Atlanta, Georgia, United States of America, 7 Department of Epidemiology, Emory University Rollins School of Public Health, Atlanta, Georgia, United States of America, 8 Department of Medicine, Albert Einstein College of Medicine and Montefiore Medical Center, Bronx, New York, United States of America, 9 Section of Endocrinology, Atlanta Veterans Affairs Medical Center, Atlanta, Georgia, United States of America

‡ These authors are co-senior authors on this work.
* jmcoll4@emory.edu

**Data Availability Statement:** All relevant data are within the manuscript and its Supporting Information files.

## Abstract

The metabolic signaling pathways that drive pathologic tissue inflammation and damage in humans with pulmonary tuberculosis (TB) are not well understood. Using combined methods in plasma high-resolution metabolomics, lipidomics and cytokine profiling from a multicohort study of humans with pulmonary TB disease, we discovered that IL-1β-mediated inflammatory signaling was closely associated with TCA cycle remodeling, characterized by accumulation of the proinflammatory metabolite succinate and decreased concentrations of the anti-inflammatory metabolite itaconate. This inflammatory metabolic response was particularly active in persons with multidrug-resistant (MDR)-TB that received at least 2 months of ineffective treatment and was only reversed after 1 year of appropriate anti-TB chemotherapy. Both succinate and IL-1β were significantly associated with proinflammatory lipid signaling, including increases in the products of phospholipase A2, increased arachidonic acid formation, and metabolism of arachidonic acid to proinflammatory eicosanoids. Together, these results indicate that decreased itaconate and accumulation of succinate and other TCA cycle intermediates is associated with IL-1β-mediated proinflammatory eicosanoid signaling in pulmonary TB disease. These findings support host metabolic remodeling as a key driver of pathologic inflammation in human TB disease.

**Funding:** The study has received funding from: National Institutes of Health grant R21 AI130918 (TRZ, NRG, JCMB, DPJ, RRK, HMB) National Institutes of Health grant K23 AI144040 (JMC) National Institutes of Health grant P30 AI050409 (JMC) National Institutes of Health grant T32 AI074492 (JMC) National Institutes of Health grant P30 ES019776 (DPJ, TRZ) National Institutes of Health grant R01 AI087465 (NRG, NSS, JCMB) National Institutes of Health grant K24 AI114444 (NRG) National Institutes of Health grant D43 TW007124 (RRK, HMB) National Institutes of Health grant R01AI114304 (JCMB) National Institutes of Health grant R01AI145679 (JCMB) National Institutes of Health grant K24AI155045 (JCMB) National Institutes of Health grant P30AI124414 (JCMB) National Institutes of Health grant UL1 TR002378 (JMC, MK, HMB, EAO, TRZ) National Institutes of Health grant UL1TR001073 (JCMB) Emory University Global Health Institute (TRZ, HMB, NT, RRK) Emory Medical Care Foundation (RRK, JMC, HMB, TRZ) The funders had no role in study design, data collection and analysis, decision to publish, or preparation of the manuscript.

**Competing interests:** The authors have declared that no competing interests exist.

## Author summary

Pulmonary tuberculosis (TB) often results in pathologic lung inflammation that causes tissue damage and does not control bacterial replication. This impairs the host response to antibiotic treatment and can result in long term deficits in lung function. Currently, the role of host metabolism in regulating the inflammatory response in TB disease is not well understood. Here, we use detailed immunometabolic phenotyping to show that metabolic remodeling of the tricarboxylic acid (TCA) cycle is closely associated with pathologic inflammatory signaling in humans with TB disease. Accumulation of TCA cycle intermediates in plasma, including the proinflammatory metabolite succinate, as well as decreased concentrations of the anti-inflammatory metabolite itaconate, were associated with increases in IL-1β and upregulation of proinflammatory lipid signaling cascades. This inflammatory network was upregulated following delays in appropriate anti-TB treatment and was associated with prolonged time to sputum culture clearance of *Mycobacterium tuberculosis*. Our study provides new insights into the metabolic reprograming that leads to pathologic inflammation in humans with pulmonary TB.

## Introduction

Prior to the COVID-19 pandemic, tuberculosis (TB) was the leading global cause of infectious disease mortality, accounting for ~1.4 million deaths each year [1]. In persons successfully treated for pulmonary TB, rates of post-TB obstructive lung disease are high [2,3] and may be even higher in persons treated for multidrug resistant (MDR)-TB [4]. Such long-term changes in pulmonary function are likely the result of pathologic lung inflammation and tissue damage prior to and shortly after initiation of anti-TB chemotherapy [5]. Elucidating host response pathways that promote pathologic tissue inflammation in persons with pulmonary TB will be critical to detect those at greatest risk for adverse pulmonary outcomes and identify targets for host-directed therapeutics.

The cytokine IL-1 plays a complex role following infection with *Mycobacterium tuberculosis* (*Mtb*) that appears to evolve as one progresses to TB disease. In mice infected with *Mtb*, IL-1 initially augments production of prostaglandin E2 (PGE2) and limits production of type I interferons, thereby enhancing control of bacterial replication [6]. However, in later stages of TB disease, IL-1 signaling leads to upregulation of proinflammatory eicosanoids and an influx of neutrophils, which promote further bacterial replication and tissue destruction [7]. Similarly, IL-1 signaling in macaques with TB disease is highly correlated with pulmonary inflammation, and IL-1 receptor blockade attenuates tissue damage [8]. This suggests that as TB disease progresses, IL-1 signaling becomes pathologic and leads to tissue inflammation that is permissive to bacterial growth. The primary drivers of IL-1 signaling in humans with pulmonary TB and whether such signaling yields a primarily proinflammatory or anti-inflammatory phenotype has yet to be established.

Recent studies in macrophage biology show tricarboxylic acid (TCA) cycle remodeling may play an important role in regulating IL-1 [9–11]. Inflammatory macrophage activation leads to accumulation of TCA cycle intermediates such as succinate, which, in turn, results in upregulation of the proinflammatory IL-1β-HIF-1α axis [9–11]. The metabolite itaconate counteracts this inflammatory metabolic remodeling by inhibiting succinate dehydrogenase and stabilizing the anti-inflammatory transcription factor Nrf2 [9,12]. However, the contribution of this host metabolic response pathway to inflammation in human pulmonary TB has not been previously described.

To better characterize the contribution of host metabolism to pathologic inflammatory cascades in humans with pulmonary TB, we performed detailed immunometabolic profiling using combined approaches in targeted and untargeted high-resolution metabolomics (HRM), lipidomics and cytokine profiling in a multicohort study of persons with drug susceptible (DS) and MDR pulmonary TB. Our aim was to gain a more comprehensive understanding of the metabolic drivers of inflammation in human TB disease. Here we show TCA cycle remodeling, characterized by accumulation of TCA cycle intermediates and decreased itaconate, is closely associated with molecular response networks of pathologic inflammation in human TB disease.

## Results

### Metabolic Pathway Regulation in MDR-TB

We hypothesized that persons experiencing delays in the receipt of effective anti-TB chemotherapy would experience continued progression of pulmonary TB disease and propagation of proinflammatory signaling cascades. We therefore sought to determine which metabolic pathways were most regulated in a population with MDR-TB (n = 85) that was initially treated with at least 2 months of ineffective, first-line anti-TB drug therapy prior to the diagnosis of MDR-TB (Table 1; described in Methods) [13]. Persons with MDR-TB were enrolled from 2011–2013 and prior to widespread implementation of rapid molecular diagnostic testing (Xpert MTB/RIF). The diagnosis of MDR-TB was made after a positive sputum culture for *Mtb* demonstrated MDR-TB using phenotypic drug susceptibility testing (DST). We compared the plasma metabolome in the MDR-TB cohort to a population of persons with drug susceptible (DS)-TB (n = 89) enrolled within 1 week of pulmonary TB diagnosis as well as a group of control participants, which included asymptomatic individuals with LTBI (n = 20)

**Table 1. Clinical and demographic characteristics of study participants.**

| Participant Characteristics | MDR-TB HIV positive (n = 64) | MDR-TB HIV negative (n = 21) | DS-TB (n = 89) | LTBI (n = 20) | Controls without Mtb infection (n = 37) |
|---|---|---|---|---|---|
| Female sex, n (%) | 42 (66) | 11 (52) | 31 (35) | 9 (45) | 31 (84) |
| Age, years (median [IQR]) | 35 (29–41) | 28 (20–48) | 31 (24–43) | 39 (28–44) | 51 (41–56) |
| CD4, cells/mm$^3$ (median [IQR]) | 229 (138–400) | N/A | N/A | N/A | N/A |
| HIV Viral Load, copies/mL (median [IQR]) | 122 ($<$40–14,850) | N/A | N/A | N/A | N/A |
| Receiving anti-retroviral therapy at enrollment n (%) | 50 (78) | N/A | N/A | N/A | N/A |
| TB disease history, n (%) | | | | N/A | N/A |
| No TB history | 21 (33) | 9 (43) | 89 (100) | | |
| Yes, completed treatment | 31 (48) | 6 (29) | | | |
| Yes, failed treatment | 12 (19) | 6 (29) | | | |
| +AFB smear at first study visit, n (%)* | 28 (56) | 8 (50) | 67 (75) | N/A | N/A |
| +Sputum culture at diagnosis, n (%) | 64 (100) | 21 (100) | 89 (100) | N/A | N/A |
| +Sputum culture at first study visit, n (%)** | 56 (79) | 16 (80) | 76 (85) | N/A | N/A |
| Time to sputum culture conversion, days (median [IQR]) § | 81 (51–90) | 57 (53–80) | 28 (22–47) | N/A | N/A |

Multidrug resistant (MDR); Drug susceptible (DS)

*AFB sputum smear results were not available for 19 MDR-TB participants (22.4%)

**AFB sputum culture results were not available for 14 MDR-TB participants (16.5%)

§Data on sputum culture conversion was missing for 7 MDR-TB participants (12.5%) with a positive sputum culture at baseline

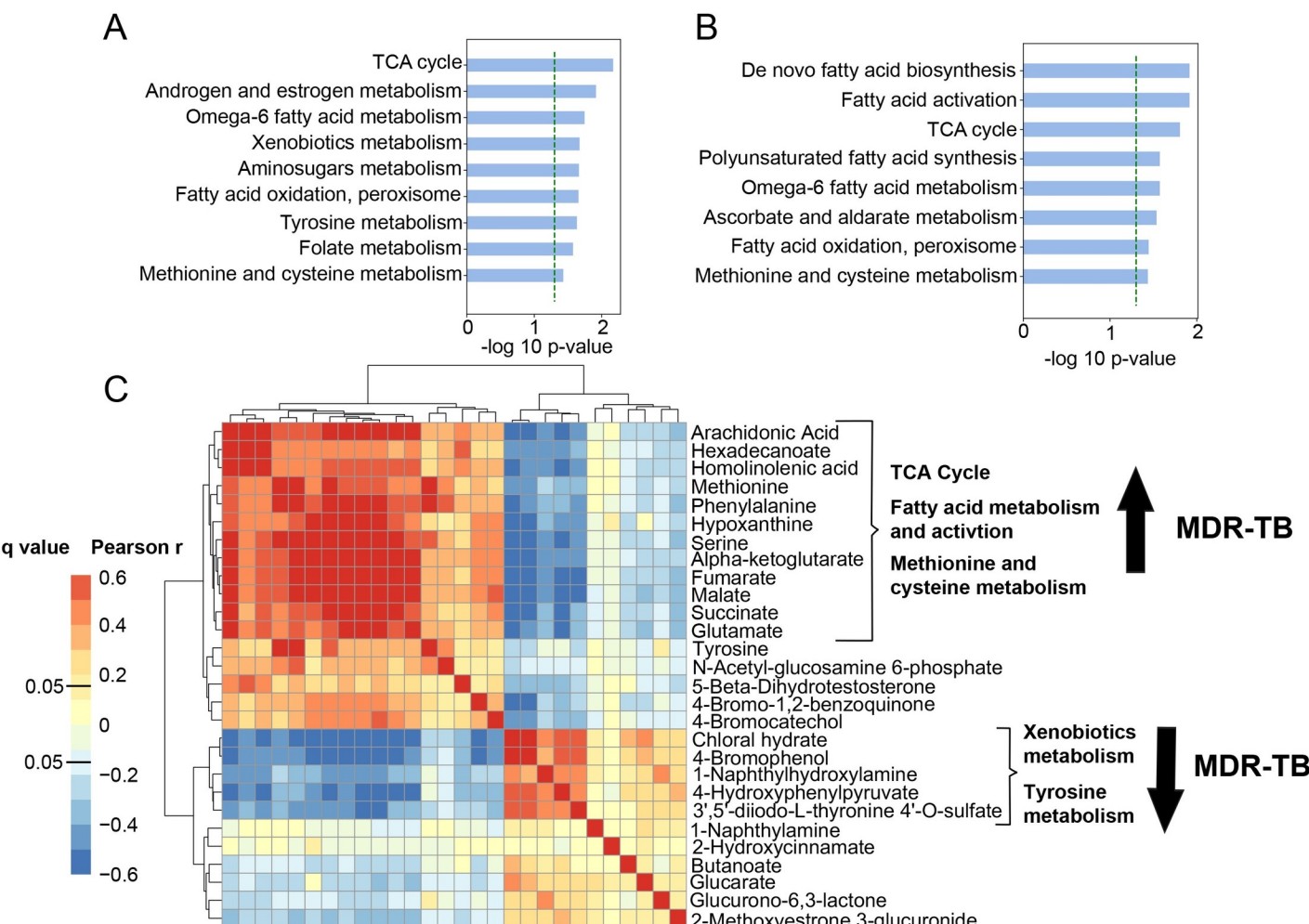

**Fig 1. Metabolic pathway regulation in MDR-TB.** (A) Unbiased pathway analysis using plasma HRM with C18 negative chromatography to compare the metabolome of persons with multidrug resistant (MDR)-TB (n = 85) to (A) asymptomatic controls without TB disease (n = 57) and (B) persons with drug susceptible (DS)-TB (n = 89) shows TCA cycle metabolism and multiple overlapping pathways in fatty acid synthesis, activation and metabolism were the most significantly regulated pathways in MDR-TB. The y-axis shows significantly regulated metabolic pathways and the x-axis shows the -log p-value for pathway enrichment adjusted for age, sex and HIV status. (C) Correlation analysis of primary metabolite adducts (M-H) mapping to significant pathways reveals a network of highly correlated metabolites involved in the TCA cycle, fatty acid metabolism and methionine and cysteine metabolism that were significantly increased in persons with MDR-TB.

enrolled from a clinic serving refugees recently arrived in the U.S. and a population of U.S.-born individuals without evidence of TB disease or *Mtb* infection (n = 37) [14].

With plasma HRM, we detected 9,787 m/z features using C18 liquid chromatography in negative ionization mode, of which 5,146 were present in >90% of samples and selected for downstream analysis. We first sought to determine which metabolic pathways were significantly regulated in persons with MDR-TB versus controls and persons with DS-TB. After adjustment for age, sex and HIV status, unbiased pathway analysis [15] revealed the TCA cycle and multiple overlapping pathways in fatty acid metabolism, including de novo fatty acid biosynthesis, fatty acid activation, and fatty acid oxidation, were the most significantly regulated metabolic pathways in persons with MDR-TB versus persons without TB disease (Fig 1A) as well as persons with DS-TB (Fig 1B). We then performed a correlation analysis of metabolites with primary adducts (M-H) mapping to significant metabolic pathways. This analysis yielded a cluster of highly correlated metabolites that were upregulated in the MDR-TB cohort including arachidonic acid,

homolinoleic acid and TCA cycle intermediates alpha-ketoglutarate, succinate, fumarate and malate (p<0.001 for all; Fig 1C). There was a smaller cluster of correlated metabolites mapping to the xenobiotics and tyrosine metabolic pathways that were down-regulated in MDR-TB.

Accumulation of succinate and downstream TCA cycle intermediates has been described as a potent driver of inflammation in multiple disease states [9,10,16]. We therefore hypothesized that TCA cycle remodeling in MDR-TB represented a proinflammatory signaling cascade. To better characterize observed changes in this metabolic pathway, TCA cycle intermediates were confirmed and quantified by accurate mass, MS/MS and retention time relative to authentic standards [17]. TCA cycle remodeling in persons with MDR-TB both with and without HIV coinfection was characterized by significant increases in plasma concentrations of succinate, fumarate, and malate versus persons with DS-TB, LTBI and controls without evidence of infection with *Mtb* (p<0.001 for all; Fig 2). Persons with MDR-TB also exhibited significant increases in plasma concentrations of alpha-ketoglutarate and glutamate (p<0.001 for all) whereas concentrations of citrate were significantly decreased in persons with MDR-TB and concentrations of cis-aconitate were similar between groups. These findings suggest the primary source of succinate in those with MDR-TB was not flux through the TCA cycle, but rather glutamine-dependent anaplerosis to alpha-ketoglutarate as has been described in *in vitro* and mouse models [10].

The metabolite itaconate is derived from TCA cycle intermediate cis-aconitate and has been shown to attenuate inflammatory metabolic signals through inhibition of succinate dehydrogenase [9], induction of Nrf2 [12], and inhibition of inflammasome activation [18]. Thus, we posited that accumulation of TCA cycle intermediates in MDR-TB may be accompanied by decreases in itaconate. Indeed, we found itaconate was significantly decreased in HIV-positive and HIV-negative persons with MDR-TB compared to persons with DS-TB, LTBI and uninfected controls (p<0.001 for all). Plasma concentrations of itaconate were significantly and negatively correlated with plasma concentrations of succinate, fumarate, and malate (p<0.001 for all; S1A–S1C Fig) consistent with a counterregulatory role. These findings suggest persons with progressive and untreated pulmonary TB experience proinflammatory metabolic remodeling characterized by accumulation of succinate via glutamine-dependent anaplerosis and decreased itaconate.

## TCA cycle remodeling is associated with a proinflammatory plasma cytokine response

Accumulation of TCA cycle intermediates such as succinate drive inflammation by increasing production of IL-1β [9,10,16]. We therefore hypothesized that the primary role of TCA cycle remodeling seen in participants with MDR-TB was induction of IL-1β. To answer this question, we measured plasma cytokine concentrations in a subset of persons with MDR-TB (n = 37), persons with DS-TB (n = 29) and persons without *Mtb* infection (n = 20) (S1 Table). We found concentrations of IL-1β were highly correlated with plasma concentrations of TCA cycle intermediates and negatively correlated with itaconate (p<0.001 for all; Fig 3A). Using hierarchical clustering analysis (HCA) of correlations between plasma cytokines and TCA cycle intermediates, we found IL-8 and IL-4 were also highly correlated with TCA cycle intermediates and formed a cluster with IL-1β. Plasma concentrations of IL-1β were significantly elevated in HIV-positive persons with MDR-TB, HIV-negative persons with MDR-TB and persons with DS-TB versus controls without *Mtb* infection (p<0.001, p = 0.003 and p = 0.03 respectively; Fig 3B). The elevation was significantly greater in persons with MDR-TB with and without HIV-coinfection versus persons with DS-TB (p<0.001 and p = 0.03 respectively). Conversely, plasma concentrations of granulocyte-macrophage colony stimulating factor

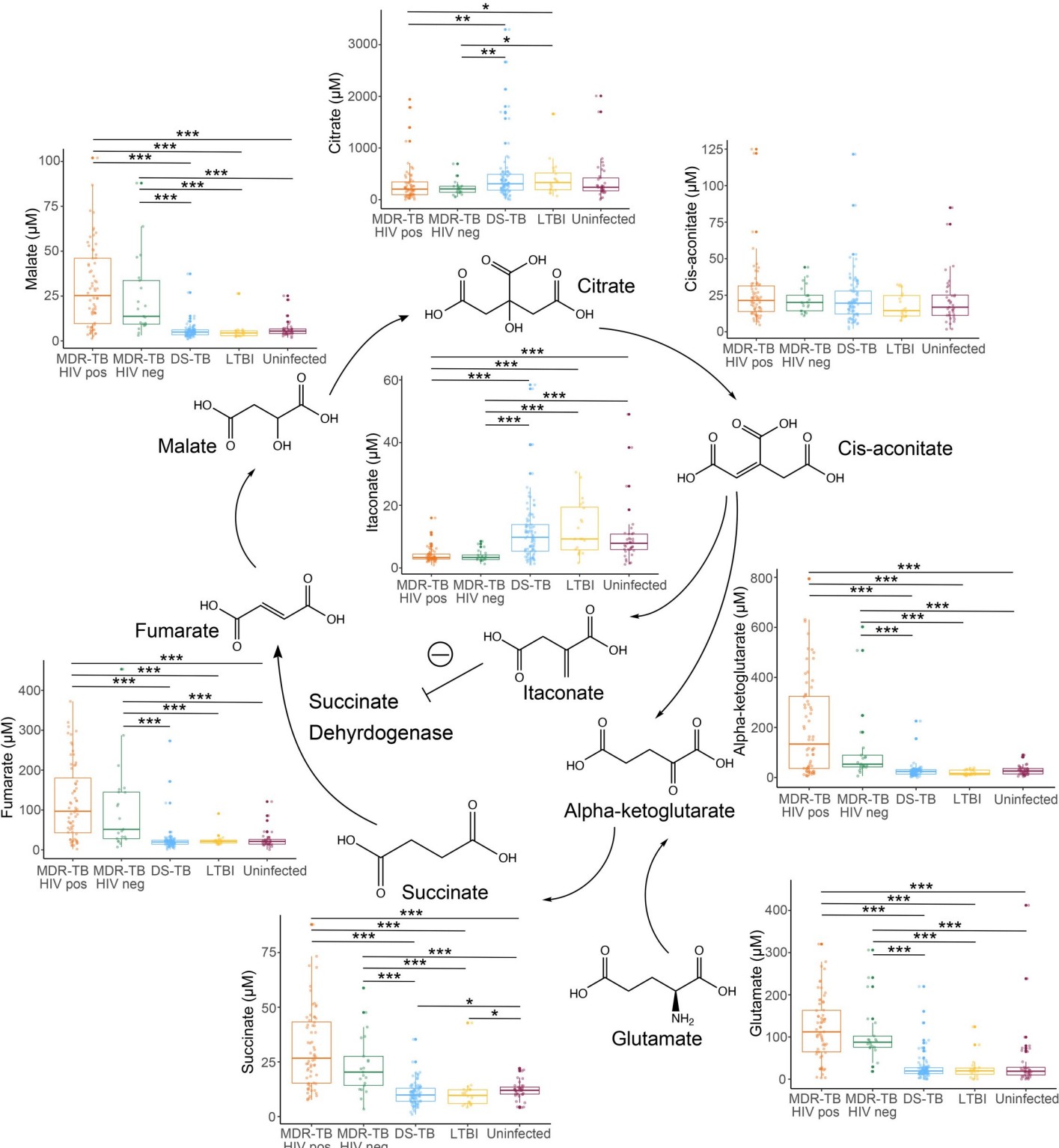

**Fig 2. Characterization of TCA cycle remodeling in MDR-TB.** Plasma quantification of TCA cycle metabolites in persons with multidrug resistant (MDR)-TB with (orange; n = 64) and without (green; n = 21) HIV co-infection shows significant increases in the metabolites alpha-ketoglutarate, succinate, fumarate, and malate versus persons with drug susceptible (DS)-TB (blue; n = 89), LTBI (yellow; n = 20), and controls without LTBI or TB disease (purple; n = 37). Plasma concentrations of glutamate were also significantly increased in MDR-TB while citrate was significantly decreased and concentrations of cis-aconitate were similar between groups. Plasma concentrations of the anti-inflammatory metabolite itaconate were significantly decreased in persons with MDR-TB. Groups were compared using the Wilcox rank sum test: * p<0.05, ** p<0.01, *** p<0.001.

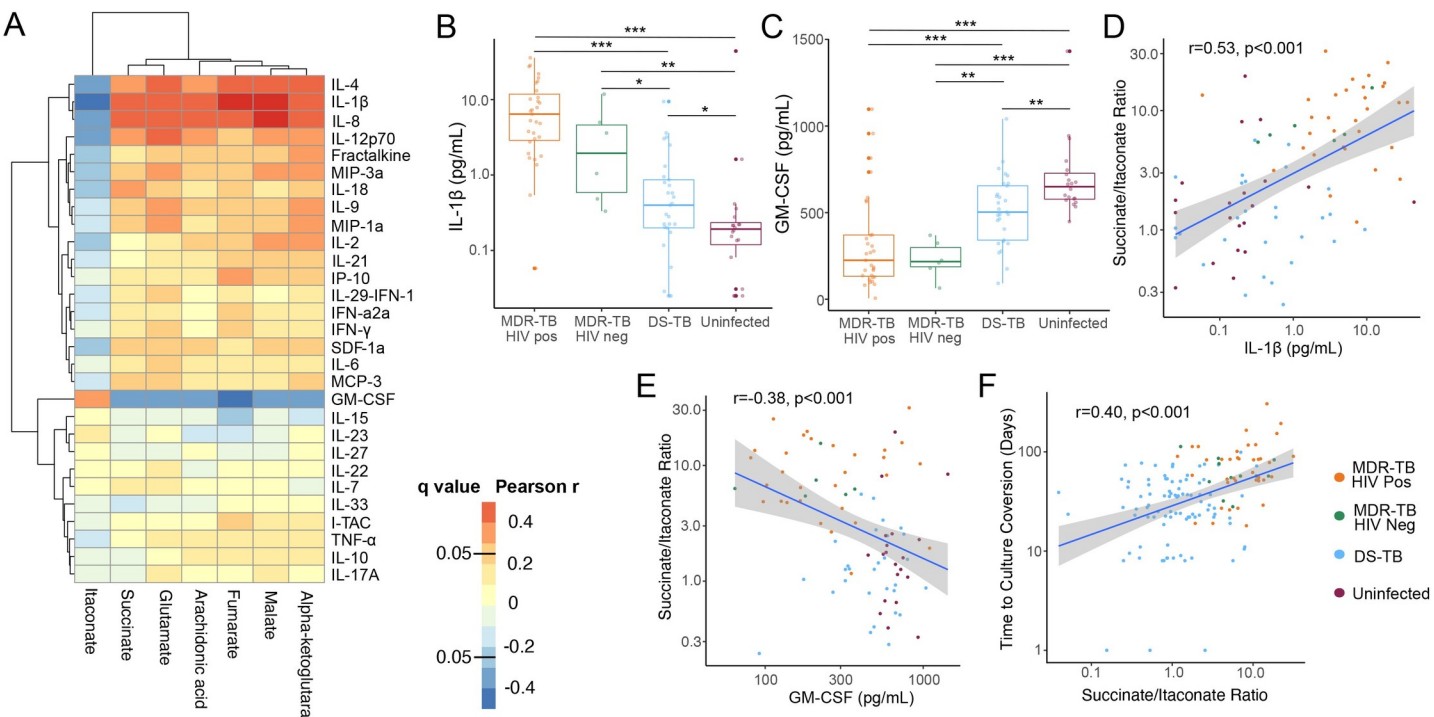

**Fig 3. TCA cycle remodeling is associated with increased IL-1β and delayed response to treatment** (A) In a subset of persons with MDR-TB with (n = 31) and without (n = 6) HIV co-infection, DS-TB (n = 29) and controls without TB infection or disease (n = 20), plasma concentrations of TCA cycle intermediates and arachidonic acid were positively correlated with concentrations of IL-1β, IL-4 and IL-8 and negatively correlated with GM-CSF. (B) Plasma concentrations of IL-1β were significantly higher in person with MDR-TB versus those with DS-TB and controls while (C) GM-CSF was significantly lower in the MDR-TB group. (D) The plasma succinate/itaconate ratio at enrolment was significantly and positively correlated with plasma concentrations of IL-1β and (E) negatively correlated with concentrations of GM-CSF. (F) The plasma ratio of succinate/itaconate was also significantly and positively correlated with subsequent time to sputum culture conversion. Groups were compared using the Wilcox rank sum test: * p<0.05, ** p<0.01, *** p<0.001.

(GM-CSF) were significantly decreased in all groups with TB disease versus controls with significantly greater decreases seen in persons with MDR-TB with or without HIV co-infection versus persons with DS-TB (p<0.001 and p = 0.002 respectively; Fig 3C). The plasma ratio of succinate to itaconate was highly correlated with the plasma concentrations of IL-1β and negatively correlated with concentrations of GM-CSF (p<0.001 for both; Fig 3D and 3E). For those participants with available culture conversion data, the plasma succinate/itaconate ratio at study enrollment was significantly and positively correlated with subsequent time to sputum culture conversion (p<0.001; Fig 3F). Time to sputum culture conversion was also positively correlated with plasma concentrations of IL-1β and negatively correlated with GM-CSF (p<0.001 and p = 0.02 respectively; S2A and S2B Fig). Additionally, persons with DS-TB and a persistently positive AFB sputum smear at study enrollment (within 1 week of treatment start) had a significantly higher plasma succinate/itaconate ratio versus those who had already converted to a negative AFB sputum smear (p = 0.04; S2C Fig). These findings indicate that TCA cycle remodeling in pulmonary TB is associated with increased inflammation through upregulation of IL-1β and down-regulation of GM-CSF, leading to greater disease severity.

## TCA cycle remodeling and secretion of IL-1β are associated with proinflammatory lipid signaling

In early *Mtb* infection, IL-1β shifts the balance of eicosanoid production toward PGE2 and away from proinflammatory eicosanoids, thereby limiting *Mtb* replication [6]. However, in

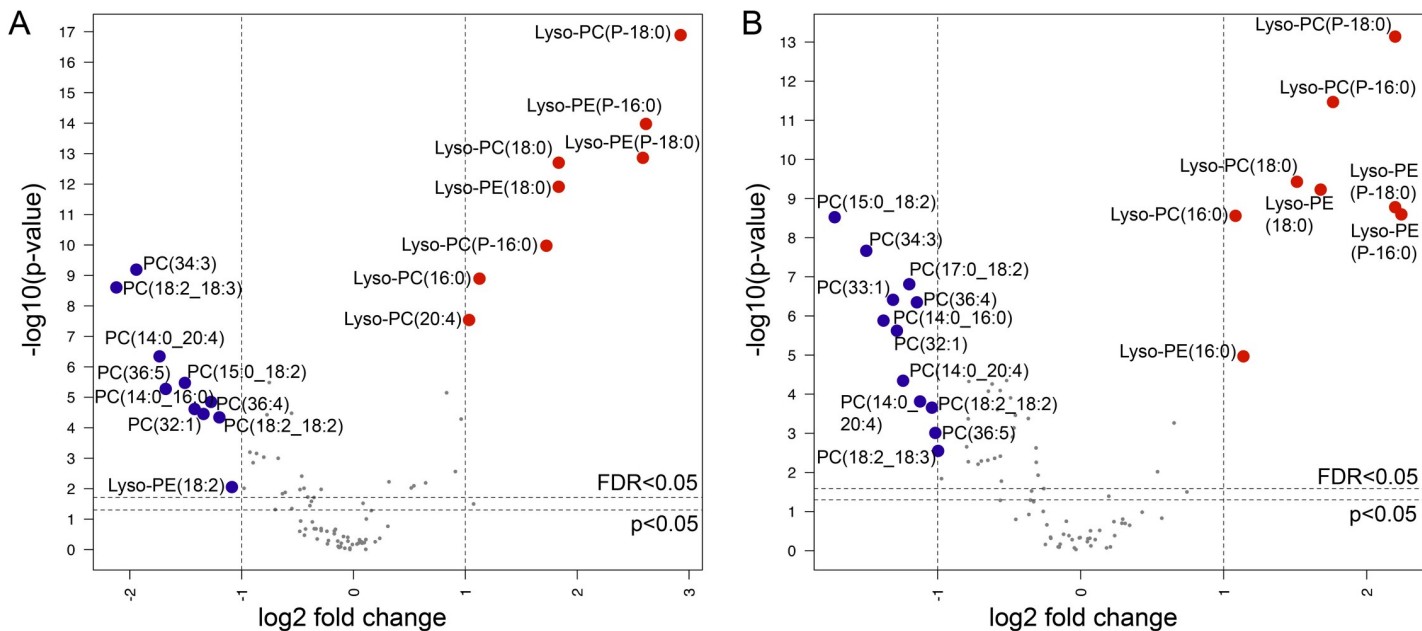

**Fig 4. Increased phospholipase activity in MDR-TB.** Volcano plots of plasma lipids with confirmed chemical identities shows significant increases of lyso-phospholipids and significant decreases in phospholipids with two acyl chains in persons with MDR-TB with (n = 29) and without (n = 21) HIV co-infection versus (A) persons with latent TB infection (n = 20) and (B) persons with drug susceptible (DS)-TB (n = 30). The x-axis shows the log2 fold change in expression in MDR-TB versus controls and DS-TB for each lipid and the y-axis shows the -log p-value for each comparison after adjustment for age, sex and HIV status. Lipids upregulated in MDR-TB with >1 log2 fold change at a false discovery rate (FDR) [60] of q<0.05 are labelled in red and those downregulated are labelled in blue.

later stages of TB disease, IL-1β signaling is associated with eicosanoid-mediated inflammation and tissue damage [7,8,19]. Given plasma concentrations of IL-1β and TCA cycle intermediates were highly correlated with concentrations of arachidonic acid, we posited that TCA cycle remodeling and IL-1β were acting primarily to drive proinflammatory lipid signaling cascades in pulmonary TB disease in humans. To examine this question, we first sought to determine whether persons with MDR-TB exhibited increased phospholipase A2 activity, the first step in formation of arachidonic acid. We performed untargeted lipidomics on plasma samples from a subset of the study population including persons with MDR-TB and HIV co-infection (n = 29), MDR-TB alone (n = 21), DS-TB (n = 30) and LTBI (n = 20) (S2 Table). Of plasma lipids with confirmed chemical identities, persons with MDR-TB demonstrated significant upregulation of multiple species of lyso-phospholipids and significant decreases in phospholipids with two acyl chains compared to persons with LTBI (Fig 4A) and persons with DS-TB (Fig 4B), consistent with significant increases in phospholipase A2 activity.

Following formation from phospholipase A2, arachidonic acid is further metabolized through one of three metabolic pathways to regulate inflammation: cyclooxygenases (COX) to form prostaglandins, lipoxygenases (LOX) to form proinflammatory eicosanoids and CYP450 enzymes to form the less biologically active dihydroxyeicosatrienoic acids (DHETs) [20]. Prior studies have shown that proinflammatory eicosanoids formed through the LOX pathway are significantly upregulated in persons with pulmonary TB disease and associated with more severe clinical disease [6,7,21] and cavity formation [22]. We therefore hypothesized that accumulation of TCA cycle intermediates, IL-1β and arachidonic acid in MDR-TB patients were part of an inflammatory cascade resulting in increased production of proinflammatory eicosanoids. To further characterize downstream changes in arachidonic acid metabolism in persons with MDR-TB versus those with DS-TB and asymptomatic controls, we performed a targeted

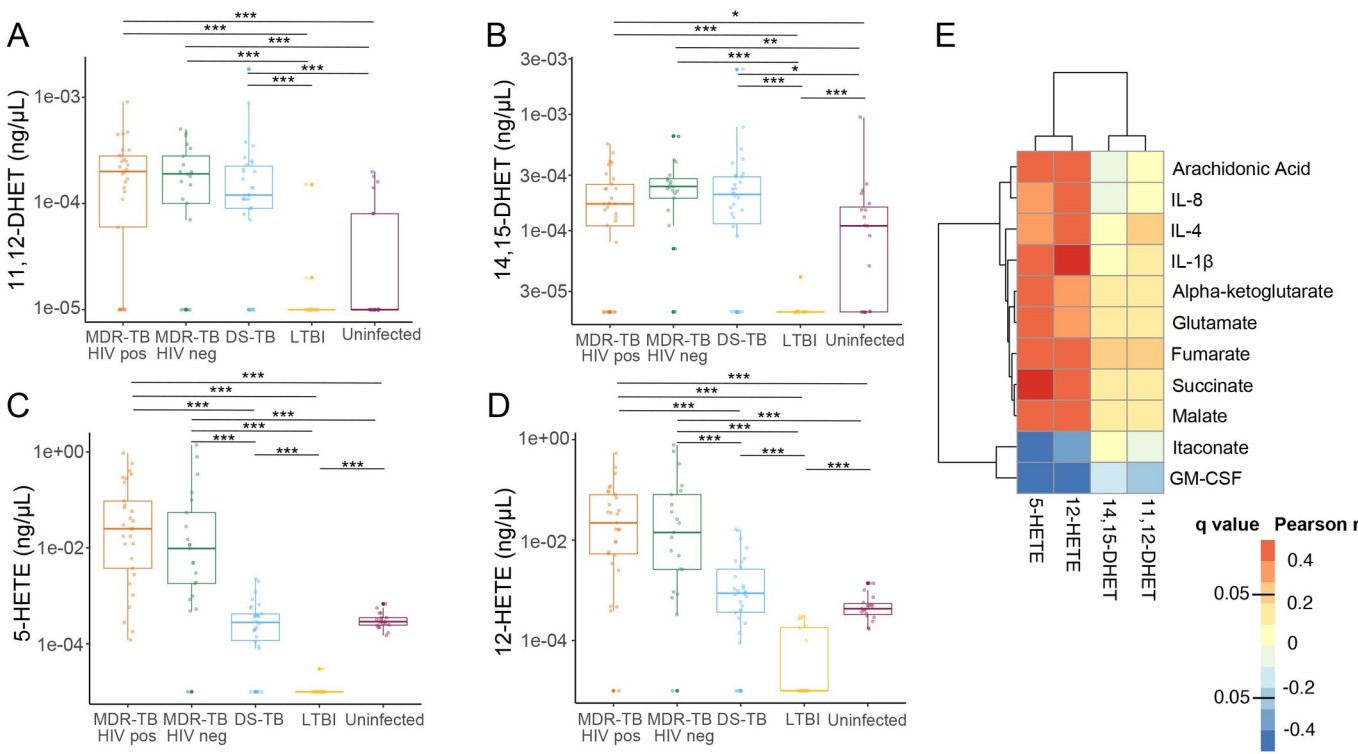

**Fig 5. Immunometabolic remodeling is associated with proinflammatory eicosanoid signaling.** Compared to persons with LTBI (yellow; n = 20) and persons without LTBI or TB disease (n = 19), persons with multidrug resistant (MDR)-TB with (orange; n = 29) and without (green; n = 21) HIV co-infection, as well as persons with drug susceptible (DS)-TB (blue; n = 30) demonstrated significant increases in plasma concentrations of the arachidonic acid metabolites (A) 11,12-DHET and (B) 14,15-DHET metabolized by the CYP450 system. Proinflammatory eicosanoids including (C) 5-HETE and (D) 12-HETE produced by lipoxygenase metabolism of arachidonic acid were significantly increased in persons with MDR-TB versus persons with DS-TB, LTBI and uninfected controls. (E) Shows a heatmap of metabolites and cytokines correlated with plasma concentrations of 5-HETE, 12-HETE, 11,12-DHET, and 14,15-DHET with dark red indicating a higher Pearson correlation coefficient and dark blue indicating a more negative correlation. Groups were compared using the Wilcox rank sum test: * p<0.05, ** p<0.01, *** p<0.001.

oxylipin assay to identify and quantify relevant eicosanoids in plasma samples [23]. All persons with pulmonary TB demonstrated significant increases in 11,12-DHET and 14,15-DHET versus controls with and without LTBI (p<0.05 for all; Fig 5A and 5B), consistent with increases in arachidonic metabolism. However, while concentrations of the less biologically active DHET molecules did not differ between TB disease groups, persons with MDR-TB with and without HIV coinfection demonstrated significant increases in the proinflammatory eicosanoids 5-HETE and 12-HETE compared to persons with DS-TB, LTBI and controls without *Mtb* infection (p<0.001 for all; Fig 5C and 5D). Similarly, leukotriene B4 was significantly increased in persons with MDR-TB with and without HIV coinfection versus other groups (p<0.001 for all) while concentrations of PGF2a were undetectable (S3 Fig). This indicates arachidonic acid metabolism in persons with MDR-TB was both increased and more likely to produce proinflammatory eicosanoids metabolized via LOX pathways compared to persons with DS-TB. Plasma concentrations of 5-HETE and 12-HETE, but not 11,12-DHETE or 14,15-DHETE, were highly correlated with plasma concentrations of succinate, fumarate and malate as well as IL-1β, IL-8 and IL-4 while they were negatively correlated with concentrations of itaconate and GM-CSF (p<0.001 for all; Fig 5E). These findings suggest TCA cycle remodeling, increased IL-1β and decreased GM-CSF are closely associated with proinflammatory lipid signaling in pulmonary TB.

## Proinflammatory metabolic remodeling is reversed with TB treatment

Because patients with MDR-TB were enrolled from a different country from those with DS-TB and controls with and without LTBI, we examined the possibility that observed differences in this population could be due to factors unrelated to TB disease. We reasoned that if TCA cycle remodeling were due to TB disease, then these changes would be reversed with appropriate anti-TB chemotherapy. We therefore examined the change in plasma concentrations of metabolites and cytokines in a subset of MDR-TB patients where plasma samples were available over the first year of treatment (n = 17). While there was minimal change in plasma concentrations of succinate, fumarate and malate at 2–4 months, there was a significant decline after 1 year of appropriate treatment for MDR-TB (p<0.05 for all; Fig 6A–6C). Plasma concentrations of these metabolites after 12–15 months of MDR-TB treatment were not significantly different than those of controls without *Mtb* infection. There was an increase in plasma itaconate concentrations after 2–4 months of treatment and 12–15 months of treatment that did not meet statistical significance (p = 0.06 and 0.12 respectively) and remained significantly lower than uninfected controls (p = 0.01; Fig 6D). Plasma concentrations of arachidonic acid also significantly declined relative to baseline after 1 year of MDR-TB treatment (p = 0.03) and were not significantly different from controls (Fig 6E). In the DS-TB group, plasma concentrations of these metabolites were unchanged after 2 months of treatment (S4 Fig). While IL-1β, IL-4 and IL-8 significantly declined after 1 year of MDR-TB treatment (p<0.05 for all), they remained significantly higher versus controls without *Mtb* infection, suggesting increases in systemic inflammation remained even after 1 year of treatment (p<0.05 for all; Fig 6F–6H). Similar to itaconate, GM-CSF did not significantly increase over the course of MDR-TB treatment and remained significantly lower than controls without *Mtb* infection (p<0.001; Fig 6I). Such ongoing immune activation could be due, in part, to concomitant administration of antiretroviral therapy in those with HIV co-infection [24], though trends of all metabolites and cytokines over time did not differ by HIV infection status in the MDR-TB group (S5 Fig). Together, these findings indicate that proinflammatory metabolic remodeling is reversed with appropriate TB treatment, but that the timeline of this reversal may be prolonged. Further, they suggest mechanisms other than itaconate and GM-CSF signaling are required to dampen inflammation.

## Discussion

In this multicohort study, we used unbiased plasma metabolic pathway analyses combined with targeted and untargeted lipidomics and cytokine profiling to discover that TCA cycle remodeling is strongly associated with IL-1β secretion and proinflammatory eicosanoid signaling in persons with pulmonary TB disease. We found this inflammatory cascade, characterized by increased plasma concentrations of TCA cycle intermediates succinate, fumarate and malate and decreased concentrations of itaconate, was particularly upregulated in persons with MDR-TB after at least 2 months of ineffective anti-TB chemotherapy and was reversed only after 1 year of efficacious treatment. Collectively, these results indicate that TCA cycle remodeling is an important driver of proinflammatory cytokine and eicosanoid signaling in pulmonary TB disease. Furthermore, our findings suggest that prolonged delays in effective treatment allow this proinflammatory cascade to perpetuate, potentially contributing to poor treatment outcomes in MDR-TB. These findings also indicate that further examination of host-directed therapies that reverse such immunometabolic remodeling is warranted [6–9,12].

Accumulation of succinate is known to drive inflammation in macrophages through stabilization of the transcription factor HIF-1α to increase transcription of *Il1b* [10,16]. This metabolic signal is the result of a shift in cellular energy metabolism toward glycolysis and away

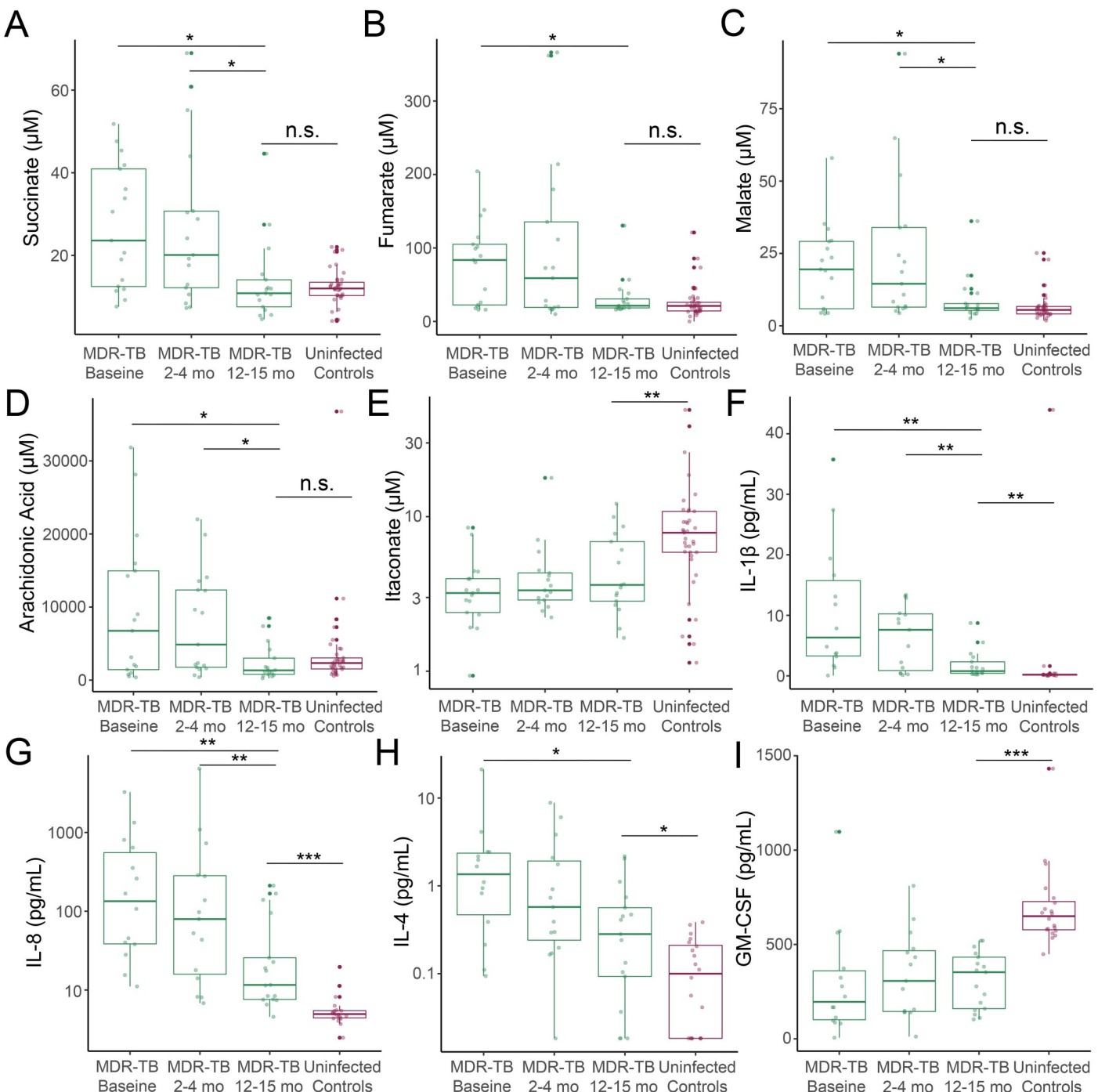

**Fig 6. Proinflammatory immunometabolic changes are reversed with appropriate TB treatment.** In persons with multidrug resistant (MDR)-TB (n = 17) here was a significant decline in plasma concentrations of (A) succinate, (B) fumarate, (C) malate, and (D) arachidonic acid after 1 year of TB therapy with concentrations declining to similar levels as controls without *Mtb* infection. (E) While plasma itaconate concentrations increased after 2–4 months and 1 year of treatment, this did not meet statistical significance and remained significantly lower than controls. Plasma concentrations of (F) IL-1β, (G) IL-8, and (H) IL-4 also significantly declined after 1 year of treatment, but remained significantly higher than uninfected controls. (I) Plasma concentrations of GM-CSF did not significantly change with MDR-TB treatment and remained significantly lower than those of controls. Groups were compared using the Wilcox signed rank test: * p<0.05, ** p<0.01, *** p<0.001.

from oxidative phosphorylation, thereby causing succinate and other TCA cycle intermediates to accumulate, a process termed aerobic glycolysis or the "Warburg Effect" [25]. Prior studies have shown *Mtb* is able to induce aerobic glycolysis in pulmonary macrophages and that this change is necessary for production of IL-1β [26,27]. The strong association between elevated succinate and IL-1β-mediated inflammatory signaling in our study suggests that TCA cycle remodeling plays a critical role in regulating the inflammatory response in humans with pulmonary TB. Further, our results indicate the primary source of TCA cycle intermediates in TB disease is glutamine-dependent anaplerosis, consistent with inflammatory metabolic remodeling driven by aerobic glycolysis [10].

However, while prior *in vitro* and mouse models have shown induction of IL-1β results in control of *Mtb* replication [6,26], we found TCA cycle remodeling and induction of IL-1β produced a distinctly proinflammatory phenotype in humans that was associated with increased bacterial burden and prolonged time to sputum culture conversion. Our findings are supported by independent human studies, which have consistently show elevated plasma concentrations of IL-1β in persons with pulmonary TB are correlated with markers of inflammation [28] and associated with greater extent of disease and cavitation on chest radiograph [19,29]. A possible explanation for these apparently discordant findings is that immunometabolic remodeling of macrophages evolves following infection with *Mtb* [30]. Initially, IL-1β provides a counterbalance to type I IFN signaling by promoting metabolism of arachidonic acid to prostaglandins [6]. Early IL-1 receptor blockade leads to greater abundance of proinflammatory eicosanoids and increased bacterial burden [6,31]. In later stages of TB disease in mice, IL-1β becomes a driver of proinflammatory eicosanoid signaling leading to an influx of neutrophilic inflammation that is permissive to bacterial growth [7]. In macaques with TB disease, plasma concentrations of IL-1β are highly correlated with levels of pulmonary inflammation and IL-1 receptor blockade limits tissue damage [8]. Thus, accumulating TCA cycle intermediates and decreased itaconate during later stages of TB disease may be another example of *Mtb* exploiting a host metabolic adaption aimed at controlling *Mtb* replication during the initial stages of infection [32,33]. While aerobic glycolysis and increased IL-1β initially promotes control of bacterial replication [26], our findings suggest that as TB disease progresses in humans, TCA cycle remodeling increases arachidonic acid metabolism and conversion to proinflammatory eicosanoids, potentially worsening tissue damage and increasing disease severity. In future studies, it will be important to determine whether such metabolic remodeling is a function of uncontrolled bacterial replication or whether the host response itself creates an environment that promotes bacterial survival.

The observation that dysregulated IL-1β signaling occurs disproportionately in persons with MDR-TB has been reported previously [28], and inhaled IFN-α, which suppresses IL-1β production in pulmonary TB disease [34] has been studied as a potential treatment for refractory MDR-TB [35]. We hypothesize that the immunometabolic remodeling in the MDR-TB group in the present study was caused primarily by propagation of inflammatory signaling cascades resulting from prior, inadequate MDR-TB treatment. However, the possibility that differences in *Mtb* strains themselves can induce differential immunometabolic phenotypes in humans warrants further investigation. The ability of *Mtb* to induce IFN-β production using the ESX-1 system, which permeabilizes the mitochondrial membrane in infected macrophages [36] and inhibits aerobic glycolysis [37], differs by strain type [38]. Further, mutations in the *Mtb rpoB* gene that confer rifampin resistance have been associated with differential metabolic responses and secretion of IFN-β and IL-1β in macrophages [39]. Thus, it remains possible that the different host responses observed in the MDR-TB group are related to differences in infecting *Mtb* strains themselves.

Our findings also demonstrate the importance of understanding the role of itaconate and GM-CSF in regulating the inflammatory response in human TB disease. Itaconate has been shown to limit inflammation in a variety of ways, including inhibition of succinate dehydrogenase [9], activation of the antioxidant transcription factor NRF2 [12] and inhibition of NRLP3 inflammasome activation [18]. *Irg1-/-* mice, which lack the enzyme that converts cis-aconitate to itaconate, experience pathologic pulmonary inflammation following infection with *Mtb* driven by an influx of neutrophils that results in rapid mortality [40]. Similarly, mice with impaired GM-CSF production fail to control *Mtb* replication [41] and GM-CSF neutralization during sub-optimal TB treatment exacerbates lung inflammation [42]. The strong negative correlation between plasma concentrations of itaconate and GM-CSF and the inflammatory eicosanoids 5-HETE and 12-HETE, which are known to stimulate neutrophilic inflammation in pulmonary TB [7], support these signaling molecules as important immune regulators in human TB disease. However, it is important to note that plasma concentrations both itaconate and GM-CSF did not significantly increase with MDR-TB treatment, suggesting alternative mechanisms are able to reverse proinflammatory metabolic signaling cascades. Further investigation into the role of these molecules in limiting tissue damage in TB disease has potential to reveal new targets for host-directed therapies.

This study is subject to several limitations. Though we demonstrate a strong association between increased plasma concentrations of TCA cycle intermediates, increased concentrations of IL-1β and proinflammatory eicosanoids, and decreased concentrations of GM-CSF and itaconate, the observational nature of the study precludes us from definitively establishing a causal relationship. While persons with MDR-TB were enrolled from three study sites, all were from the same country, so it remains possible that metabolic differences in this group are driven by host or environmental differences, such as polymorphisms in key metabolic enzymes [43] or co-infections with other endemic pathogens, rather than MDR-TB status. Finally, all measurements of metabolites, cytokines and lipids were taken from plasma samples rather than the pulmonary compartment, which was the primary site of TB disease in these cohorts. In future studies it will be important to combine systemic measurement immunometabolic signaling cascades with measures of the host microbiome [44], as well as cell subset and intracellular molecular cascade characterization data to better understand the immune cell phenotypes and signaling pathways associated with pathologic metabolic remodeling in TB disease.

In summary, we demonstrate that TCA cycle remodeling is closely associated with IL-1β-mediated proinflammatory eicosanoid signaling in humans with pulmonary TB. This remodeling is characterized by significant increases in plasma succinate concentrations and significant decreases in concentrations of itaconate. These findings provide evidence that pathologic inflammatory signaling in pulmonary TB is driven by host metabolic remodeling. Further elucidating the role of these immunometabolic networks in driving human TB disease progression will be critical to better understand differential host outcomes and identify new targets for host-directed therapies.

## Methods

### Ethics statement

All studies were approved by the Institutional Review Board (IRB) of Emory University (Atlanta, GA, USA), and by the individual IRBs associated with the original cohort studies: the Ethics Committee of the National Center for Tuberculosis and Lung Diseases of Georgia (Tbilisi, Georgia), the University of KwaZulu-Natal IRB (Durban, South Africa), and the Georgia Department of Public Health IRB (Atlanta, GA, USA), respectively, depending on the site of participant enrollment. All subjects provided written informed consent.

## Sample collection

For all cohorts, blood was collected in ethylenediaminetetraacetic acid (EDTA)-containing tubes and centrifuged; isolated plasma was immediately frozen and stored at -80˚C. Samples collected outside of the U.S. were subsequently shipped on dry ice to Emory University, Atlanta, GA, USA. All samples remained frozen during transit and were kept at -80˚C prior to metabolomics, lipidomics and cytokine analysis.

## Multidrug resistant-TB cohort

Persons with pulmonary TB from KwaZulu-Natal province, South Africa were enrolled as part of a prospective, observational cohort study of MDR-TB and HIV co-infection from 2011–2013 [13]. All persons in the South African cohort had MDR-TB as demonstrated by a positive sputum culture for *Mtb* and phenotypic DST indicating resistance to at least both isoniazid and rifampin. Baseline plasma samples were collected within 7 days of starting conventional treatment for MDR-TB. Persons with HIV-co-infection were continued on conventional anti-retroviral therapy (ART) and those not previously on ART were started on treatment. All patients were referred to a dedicated MDR-TB treatment center and treated with a standardized drug regimen that included kanamycin (15 mg/kg, maximum 1 g daily), moxifloxacin (400 mg daily), ethionamide (15–20 mg/kg, maximum 750 mg daily), terizidone (15–20 mg/ kg, maximum 750 mg daily), ethambutol (15–20 mg/kg, maximum 1200 mg daily), and pyrazinamide (20–30 mg/kg, maximum 1600 mg daily). All persons who completed the study were treated for a period of two years and serial plasma samples were analyzed at enrollment and 2–4 months and 12–15 months after treatment initiation.

## Drug susceptible-TB cohort

Persons with DS-pulmonary TB were selected from a randomized, double blind controlled trial of adjunctive high-dose cholecalciferol (vitamin $D_3$) for TB treatment conducted in the country of Georgia (clinicaltrials.gov identifier NCT00918086) [14]. All persons included in this metabolomics sub-study were HIV-negative. Inclusion criteria for patients included age ≥ 18 years and newly diagnosed active TB disease, based on a positive AFB sputum smear and confirmed by positive sputum culture for *Mtb*. Baseline plasma samples for HRM were obtained from eligible subjects within 7 days of initiating therapy with conventional dosing of first-line anti-TB drugs (isoniazid, rifampicin, pyrazinamide and ethambutol) [14]. Phenotypic DST was performed on *Mtb* isolates recovered from all persons with pulmonary TB using the absolute concentration method [45].

## Controls without active TB disease

Plasma from persons with and without LTBI was analyzed for cross-sectional comparison with pulmonary TB cases. Persons with LTBI were enrolled from the DeKalb County Board of Health in DeKalb County, GA, USA. All persons with LTBI had positive test results from at least two FDA-approved tests for LTBI (QFT, TSPOT.TB [TSPOT] and/or tuberculin skin test [TST]). All tests were interpreted according to the guidelines from the Centers for Disease Control and Prevention [46,47]. Controls without LTBI were U.S.-born adults at low risk for *Mtb* exposure and infection, who had at least one negative TST documented in medical records [48].

## Plasma metabolomics analysis

De-identified samples were randomized by a computer-generated list into blocks of 40 samples prior to transfer to the analytical laboratory where personnel were blinded to clinical and demographic data. Thawed plasma (65 μL) was treated with 130 μl acetonitrile (2:1, v/v) containing an internal isotopic standard mixture (3.5 μL/sample), as previously described [49]. The internal standard mix for quality control consisted of 14 stable isotopic chemicals covering a broad range of small molecules [49]. Samples were mixed and placed on ice for 30 min prior to centrifugation to remove protein. The resulting supernatant was transferred to low-volume autosampler vials maintained at 4˚C and analyzed in triplicate using an Orbitrap Fusion Mass Spectrometer (Thermo Scientific, San Jose, CA, USA) with c18 liquid chromatography (Higgins Analytical, Targa, Mountain View, CA, USA, 2.1 x 10 cm) with a formic acid/acetonitrile gradient. The high-resolution mass spectrometer was operated in negative electrospray ionization mode over a scan range of 85 to 1275 mass/charge (*m/z*) and stored as. Raw files [50]. Data were extracted and aligned using apLCMS [51] and xMSanalyzer [52] with each feature defined by specific *m/z* value, retention time and integrated ion intensity [50]. Three technical replicates were performed for each plasma sample and intensity values were median summarized [53].

## Metabolite identification and reference standardization

Identities of metabolites of interest were confirmed using ion dissociation methods (tandem MS/MS). Fragmentation spectra were generated using a Q Exactive HF Hybrid Quadrupole-Orbitrap Mass Spectrometer (Thermo Fisher Scientific, Waltham, MA) with parallel reaction monitoring mode using a targeted inclusion list. TCA cycle metabolites and arachidonic acid were confirmed and quantified by accurate mass, MS/MS and retention time relative to authentic standards [17].

## Untargeted lipidomics

Lipids were extracted from each plasma sample using a high throughput methyl t butyl ether (MtBE) extraction procedure with an automated and robust liquid handling instrument (Biotage Extrahera, Uppsala, Sweden) as previously described [54]. Extracted samples were dried under nitrogen and reconstituted in 200 μl 1:1 chloroform:methanol prior to injection into the LC/MS system and lipids were resolved using a Thermo Acclaim C18 reverse phase column on a Thermo Vanquish UPLC coupled to a Thermo Fusion IDX mass spectrometer (Thermo, Waltham, MA) [55]. Data were acquired at full scan mode at a resolution of 240,000 FWHM for all the samples and iterative data dependent acquisition (DDA) mode was collected on pooled samples at a resolution of 35,000 step-wise collision energy for identification of lipids. Data was processed using LipidSearch (Thermo Fisher, San Jose, CA). Lipids that contained a signal to noise ratio of greater than 10 and had a high confidence level (MS/MS) identifications with CVs less than 30% across pooled QCs were considered for downstream analysis.

## Targeted measurement of eicosanoids

For targeted analysis of oxylipins, these lipids were enriched from bulk membrane lipids by performing solid phase extraction using the Biotage Extrahera liquid handling system as previously described [23,56,57]. The resulting extracts were analyzed using a multiple reaction monitoring (MRM)-based LC/MS protocol on a QTrap 5500 (Sciex, Waltham, MA) synced to Exion LC AD system (SCIEX, Waltham, MA), whereby detected oxylipins and

endocannabinoids were fragmented and quantified against external standard curves. Missing values were imputed using half of the minimum detected value for each lipid.

### Plasma cytokine detection

The U-PLEX assay (Meso Scale MULTI-ARRAY Technology) commercially available by Meso Scale Discovery (MSD) was used for plasma cytokine detection. This technology allows the evaluation of multiplexed biomarkers by using custom made U-PLEX sandwich antibodies with a SULFO-TAG conjugated antibody and next generation of electrochemiluminescence (ECL) detection. The assay was performed according to the manufacturer's instructions (https://www.mesoscale.com/en/technical_resources/technical_literature/techncal_notes_search). In summary, 25μL of plasma from each participant was combined with the biotinylated antibody plus the assigned linker and the SULFO-TAG conjugated detection antibody; in parallel a multi-analyte calibrator standard was prepared by doing 4-fold serial dilutions. Both samples and calibrators were mixed with the Read buffer and loaded in a 10-spot U-PLEX plate, which was read by the MESO QuickPlex SQ 120. The plasma cytokines values (pg/mL) were extrapolated from the standard curve of each specific analyte.

### Statistics

Statistical comparisons of metabolite and lipid intensity values (abundance) and concentrations were performed in R version 3.6.1. For untargeted metabolomics and lipidomics analyses, metabolite intensity values were $\log_2$ transformed and compared between groups using linear regression, controlling for age, sex, and HIV status [58]. Metabolic pathway enrichment analysis was performed using *mummichog*, a Python-based informatics tool that leverages the organization of metabolic networks to predict functional changes in metabolic pathway activity [15,32,59]. Following quantification of selected metabolites and lipids, cross-sectional comparison of plasma concentrations between groups was made using the Wilcoxon Rank Sum test. Changes relative to baseline during treatment of TB disease were tested using a Wilcoxon Signed-Rank test. For correlation analyses, plasma metabolite, lipid and cytokine concentrations were normalized using log transformation. A p-value less than or equal to 0.05 was considered statistically significant. For untargeted correlation analyses, a false discovery rate (FDR) of q<0.05 was used [60].

### Supporting information

**S1 Fig. Correlation of plasma itaconate concentrations with TCA cycle intermediates.** Plasma concentrations of itaconate were significantly and negatively correlated with TCA cycle intermediates (A) succinate, (B) fumarate and (C) malate.
(TIF)

**S2 Fig. Immunometabolic remodeling and disease severity.** (A) Time to culture conversion was significantly and positively correlated with plasma concentrations of IL-1β at study enrollment and (B) negatively correlated with concentrations of GM-CSF. (C) In persons with drug susceptible (DS)-TB, the plasma ratio of succinate to itaconate was significantly increased in those with a persistently positive sputum smear for acid-fast bacilli (AFB) at enrollment versus those who converted to a negative AFB sputum smear.
(TIF)

**S3 Fig. Prostaglandin F2α and Leukotriene B4 in TB disease.** (A) Plasma concentrations of PGF2a were below the limit of detection in all TB disease groups as well as most uninfected U. S. controls while all persons with LTBI had detectable plasma concentrations. (B) LTB4 was

only detected in plasma in persons with MDR-TB.
(TIF)

**S4 Fig. Plasma concentrations of TCA cycle metabolites after 2 months of treatment for drug-susceptible TB.** Plasma concentrations of (A) succinate, (B) Malate, (C) fumarate, (D) itaconate and (E) arachidonic acid were not significantly different after 2 months of treatment versus baseline in persons with drug-susceptible TB from Georgia (n = 89).
(TIF)

**S5 Fig. Reversal of immunomedabolic remodeling with MDR-TB treatment does not differ by HIV status.** Decreases in proinflammatory metabolites and cytokines with MDR-TB treatment was similar in persons with MDR-TB with (orange; n = 12) and without (green; n = 5) HIV co-infection. In both groups, plasma concentrations of itaconate and GM-CSF were unchanged over time.
(TIF)

**S1 Table. Clinical and demographic characteristics of study participants included in the cytokine substudy.**
(DOCX)

**S2 Table. Clinical and demographic characteristics of study participants included in the lipidomics substudy.**
(DOCX)

**S1 Data. Untargeted plasma metabolomics feature table.**
(TXT)

**S2 Data. Table of plasma concentrations of targeted metabolites.**
(TXT)

**S3 Data. Table of plasma cytokine concentrations.**
(TXT)

**S4 Data. Untargeted plasma lipidomics feature table.**
(TXT)

**S5 Data. Table of plasma oxylipin concentrations.**
(TXT)

**S6 Data. Clinical and demographic characteristics of study participants.**
(TXT)

## Acknowledgments

We thank all the study teams and study sites from the multiple cohorts leveraged by this metabolomics study, including the Georgia National Center for Tuberculosis and Lung Disease, the University of KwaZulu-Natal and the DeKalb County Board of Health, Georgia, USA.

## Author Contributions

**Conceptualization:** Jeffrey M. Collins, Dean P. Jones, Ashish Sharma, James C. M. Brust, Rafick-Pierre Sékaly, Neel R. Gandhi, Henry M. Blumberg, Eric A. Ortlund, Thomas R. Ziegler.

**Data curation:** Jeffrey M. Collins, Dean P. Jones, Ashish Sharma, Manoj Khadka, Ken H. Liu, Kristal Maner-Smith.

**Formal analysis:** Jeffrey M. Collins, Ashish Sharma, Manoj Khadka, Kristal Maner-Smith.

**Funding acquisition:** Jeffrey M. Collins, Dean P. Jones, Russell R. Kempker, Nestani Tukvadze, N. Sarita Shah, James C. M. Brust, Neel R. Gandhi, Henry M. Blumberg, Eric A. Ortlund, Thomas R. Ziegler.

**Investigation:** Jeffrey M. Collins, Dean P. Jones, Ashish Sharma, Manoj Khadka, Ken H. Liu, Brendan Prideaux, Kristal Maner-Smith, Neel R. Gandhi, Henry M. Blumberg, Thomas R. Ziegler.

**Methodology:** Jeffrey M. Collins, Dean P. Jones, Ashish Sharma, Manoj Khadka, Ken H. Liu, Kristal Maner-Smith, Eric A. Ortlund, Thomas R. Ziegler.

**Project administration:** Jeffrey M. Collins, Russell R. Kempker, Nestani Tukvadze, N. Sarita Shah, James C. M. Brust, Henry M. Blumberg, Eric A. Ortlund, Thomas R. Ziegler.

**Resources:** Jeffrey M. Collins, Rafick-Pierre Sékaly, Thomas R. Ziegler.

**Software:** Manoj Khadka, Ken H. Liu.

**Supervision:** Dean P. Jones, Russell R. Kempker, James C. M. Brust, Rafick-Pierre Sékaly, Neel R. Gandhi, Henry M. Blumberg, Eric A. Ortlund, Thomas R. Ziegler.

**Validation:** Jeffrey M. Collins, Ashish Sharma, Manoj Khadka, Kristal Maner-Smith, Eric A. Ortlund.

**Visualization:** Jeffrey M. Collins, Ashish Sharma, Ken H. Liu, Brendan Prideaux, Rafick-Pierre Sékaly, Thomas R. Ziegler.

**Writing – original draft:** Jeffrey M. Collins.

**Writing – review & editing:** Jeffrey M. Collins, Dean P. Jones, Ashish Sharma, Manoj Khadka, Ken H. Liu, Russell R. Kempker, Brendan Prideaux, Kristal Maner-Smith, Nestani Tukvadze, N. Sarita Shah, James C. M. Brust, Rafick-Pierre Sékaly, Neel R. Gandhi, Henry M. Blumberg, Eric A. Ortlund, Thomas R. Ziegler.

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
