## [Decision Letter · Decision Letter 0]

7 Jun 2021

Dear Dr. Collins,

Thank you very much for submitting your manuscript "TCA cycle remodeling drives proinflammatory signaling in humans with pulmonary tuberculosis" for consideration at PLOS Pathogens. As with all papers reviewed by the journal, your manuscript was reviewed by members of the editorial board and by several independent reviewers. In light of the reviews (below this email), we would like to invite the resubmission of a significantly-revised version that takes into account the reviewers' comments.

We cannot make any decision about publication until we have seen the revised manuscript and your response to the reviewers' comments. Your revised manuscript is also likely to be sent to reviewers for further evaluation.

Sincerely,

David M. Lewinsohn

Associate Editor

PLOS Pathogens

JoAnne Flynn

Section Editor

PLOS Pathogens

Kasturi Haldar

Editor-in-Chief

PLOS Pathogens

orcid.org/0000-0001-5065-158X

Michael Malim

Editor-in-Chief

PLOS Pathogens

orcid.org/0000-0002-7699-2064

Reviewer's Responses to Questions

**Part I - Summary**

Reviewer #1: The manuscript by Collins et al., investigated the role of host metabolism in regulating the inflammatory response to tuberculosis. Using a combination of metabolomics, lipidomics and cytokine profiling on a multicohort study of humans with pulmonary TB, the authors demonstrated that MDR-TB disease is associated with a switch from anti-inflammatory metabolite itaconate to inflammatory metabolite succinate, leading to increased production of IL1 beta. In addition, the authors showed that this metabolic switch was associated with an increased production of inflammatory eicosanoids for the lipoxygenase pathways. Furthermore, the authors show that this metabolic switch could be reversed following 12-month anti-TB treatment. The effect of MDR Tb strains on metabolic reprogramming and subsequent immune response represents an important avenue of investigations and this study provides new evidence that MDR-TB infection has unique effect on host metabolism. The study is clearly written and logically presented, but further work and clarification are needed to fully support the findings.

Reviewer #2: In this paper the authors aim to demonstrate how pulmonary TB disease leads to dysregulation of IL-1B signalling, through disruption of the TCA cycle – leading to downstream changes in inflammatory lipid signalling. They discuss how MDR-TB patients demonstrate more disruption, inflammation and therefore tissue damage than their DS or uninfected controls. They go on to explain how these effects are reversed on effective treatment.

Methods

This is an ex vivo study, where conclusions about disease of the pulmonary compartment are drawn from analysis of plasma from peripheral blood sampling. Acellular work from a body compartment remote from the source of disease, makes it difficult to ascertain whether the two are linked. Results are therefore generally representative of correlations, and not causation (this is acknowledged by the authors in the discussion).

Reviewer #3: In this manuscript by Collins et al the authors profile the metabolome of patients with MDR-TB and compare it to uninfected controls and Drug Sensitive-TB patients. They identify strong correlations between an accumulation of TCA intermediates, IL1b and pro-inflammatory lipid mediators suggesting a metabolic reprogramming in active TB patients that are not responding to antibiotic therapy. A comparison of these changes over time show reversal of several metabolites suggesting these changes are associated with progressive disease. The authors suggest that targeting this immunometabolic reprogramming may lead to potential host-directed therapy to prevent TB. This study is a nice catalog of systemic metabolic changes that is robust and well controlled. Understanding factors associated with progressive disease is very important and will be of interest to many readers. Limitations of the study are the descriptive nature relying on correlations and thus providing limited mechanistic insight, and no truly novel discoveries (ie most of the findings were previously shown in other systems). However, the metabolic links connect several disparate pieces of data to formulate a more complete model which is significant and novel. Specific comments are below:

**Part II – Major Issues: Key Experiments Required for Acceptance**

Reviewer #1: 1) The current study highlights important differences between MDR TB-cohort and DS-TB cohorts/controls. However, the high prevalence of HIV co-infection limits the enthusiasm of the reviewer. Given the known effect of viral infection on host metabolism and eicosanoid production, the co-infection with HIV could lead to under- or over-estimation of the response. The authors should consider separating the MDR-TB cohort into HIV+ and HIV- throughout the study.

2) In addition, the control groups were not clear as it appears to be a mix of LTBI to healthy individuals. I would suggest the authors to separate the LTBI from Healthy controls for the analysis

2) In Figure 3, the data shows that several cytokines are correlated with TCA cycle intermediates/succinate. Why did the authors choose to focus on IL1-beta, considering the existent literature that MDR limits IL1beta production by inducing IFN-I (Howard et al., 2018, Nat Microbiol)? For example, GM-CSF levels are negatively correlated with succinate. Given the predominant role of GM-CSF in combination with antibiotic treatment (Benmerzoug et al., 2018, Scientific Reports), the authors should perform the same experiments as in Figure 3B-D with IL1-beta.

3) In Figure 5, please show a heatmap or correlation between the different oxylipins and TCA metabolites as in Figure 3. In addition, the authors demonstrate that infection with MDR is associated with a shift towards increased production of lipoxygenase derived eicosanoids, such as 5-HETE and 12-HETE. What is the effect of MDR on prostanoids (e.g. PGE2) or lipoxins (e.g. lipoxin A4) which have been previously linked to susceptibility/resistance to TB (PMID: 20622882, PMID: 18955568, PMID: 24990750 ). Is there any correlation between these particular eicosanoids and GM-CSF?

Reviewer #2: Study groups

This study has been performed on a multi-cohort group of patients who were recruited for other studies. The cohort of MDR-TB patients were recruited from a previous study designed to assess treatment outcomes for HIV+/MDR patients vs HIV-/MDR patients. The cohort for the study under review has a 75% positivity rate for HIV infection – we are not given any information on the CD4 counts, viral load titres, treatment regimen (if any), treatment duration etc. These patients were recruited before the introduction of WHO recommendations to treat people diagnosed with HIV at diagnosis, and would therefore likely have been subject to previous guidelines where HIV treatment was based on CD4 counts. These patients have likely therefore had a period of unsuppressed HIV infection, before having a CD4 count low enough to merit therapy. Considering there are studies looking at the effects of HIV on cellular metabolism, this is a highly relevant confounding factor on their study selection. The authors sate that the reported changes are not due to HIV.

Other important features of this study group (by review of reference 13) are that the HIV positive cohort are 70% female and over 70% have been previously treated for TB, compared to 46% female and 46% previously treated for TB in the HIV- group. It’s unclear how these baseline characteristics carried over to the selected cohort for the current study.

In relation to the DS and control groups: both of these groups differ in country of origin to the MDR group. A study in PNAS looked at differences in the crystal structure of the itaconic acid pathway enzymes (https://doi.org/10.1073/pnas.1908770116) and found major differences between people of different ethnicity, specifically a higher rate of a mutation related to enzyme activity among African ethnicity compared to American, Asian and European. Comparing differences in this immune pathway across groups of different ethnicity is therefore challenging.

Rates of concurrent (infectious) illness, or of previous exposure is not discussed, and most likely varies between the groups – considering the significant differences in disease epidemiology between South Africa and the United States.

Lastly, the Control group seems to consist of people with LTBI and those without – why are these not separated? This makes it hard to disentangle differences between those with LTBI, those with exposure and no LTBI, and those without prior exposure.

Sample methods/analysis

The authors have examined plasma samples taken from patients over the length of a treatment course for MDR/DS-TB. Each of these cohorts will be on differing drug regimens, with different administration frequencies, with potentially individual effects on cellular metabolism. This extends further to the HIV+ cohort, who, based on the original cohort study information, are mostly also on ART during the study period. Any changes in glycolytic/oxidative metabolism caused by drug therapy might lead to changes in TCA cycling, and therefore metabolite production/signalling. Furthermore, sampling the MDR-TB and DS-TB groups at 12-15 months presumably means one groups has finished drug therapy and the other not. The control group is also not on any drug therapy, at any point. This has not been acknowledged.

The other issue with sample analysis is to draw conclusions about disease in the pulmonary compartment by studying plasma samples. Bronchoalveolar lavage samples have not been studied. As the authors state themselves, the microbiome contributes significantly to succinate presence in the plasma. This study seemed to be limited by samples that had been previously collected during the original MDR-TB treatment study (reference 13).

Assessment of lipidomics is performed on a “subset” of the original population – we are given no information, however, of the rate of HIV infection or other baseline characteristics in this group. This is also the case for the cytokine analysis (Fig 6).

The study is limited by lack of mechanistic analysis. It is acknowledged by the authors that the results are observational, however they draw a bold conclusion that the IL-1B/eicosanoid signalling pathway could be targeted by HDT: I would feel it is difficult to jump to this conclusion with observational data, particularly in a pathway this novel. The relationship between itaconate and Mtb in humans is somewhat unexplored in even in vitro studies – one would imagine mechanistic interrogation would be quite important to understand given the various roles of itaconate that we currently know about (inhibition of SDH, stabilisation of NRF2, inhibition of isocitrate lyase).

Reviewer #3: 1. One shortcoming of this paper is the lack of follow up studies to test the mechanisms and the model presented. While this is understandable given the human patients, it does raise questions that are important to consider. For example the authors cannot distinguish whether the increased inflammation creates an environment that drives bacterial replication or if uncontrolled bacterial replication is driving inflammation. This is not entirely addressable by the nice inclusion of drug treatment conditions but could be addressed more explicitly in the discussion.

2. Do the authors think the metabolic reprogramming is solely driven by the host response, or could factors encoded by Mtb also contribute to this process. For example, the ESX-1 secretion system is known to disrupt mitochondria would this accelerate the switch to glycolysis?

3. Did the authors examine IL1Ra which was shown in the mouse model to also drive disease and over-ride elevated IL1b and shutting down IL1R signaling?

4. The fact that Itaconate is not significantly different following 12 months of treatment should be discussed more thoroughly as it might suggest alternative mechanisms to dampen inflammation.

5. Given IL1b is a key part to the story, are the authors able to examine whether signal 1 (Pro-IL1b) and/or signal 2 (inflammasome activation) are different based on their samples in hand? This may help drive at some mechanism given the constraints of these samples. If not maybe discuss these possibilities and what could drive each (ie more Mtb more TLR activation/Signal 1 or more cell damage so more inflammasome activation).

**Part III – Minor Issues: Editorial and Data Presentation Modifications**

Reviewer #1: 1) In general, is there any correlation between the metabolic and lipidomic reprogramming observed with MDR-TB and circulating levels of immune cells in comparison to DS-TB or controls? What is the level of protection from the macrophages derived from MDR-TB individuals compared to DS-TB?

2) Line 792: Immonometabolic should be immunometabolic

3) In Figure 6, Please show the controls on the same graphs demonstrating the return to baseline following TB treatment.

Reviewer #2: (No Response)

Reviewer #3: (No Response)

PLOS authors have the option to publish the peer review history of their article (what does this mean?). If published, this will include your full peer review and any attached files.

Reviewer #1: No

Reviewer #2: No

Reviewer #3: No
---

## [Decision Letter · Decision Letter 1]

3 Sep 2021

Dear Dr. Collins,

We are pleased to inform you that your manuscript 'TCA cycle remodeling drives proinflammatory signaling in humans with pulmonary tuberculosis' has been provisionally accepted for publication in PLOS Pathogens.

Before your manuscript can be formally accepted you will need to complete some formatting changes as well as addressing the minor comment from one reviewer, which you will receive in a follow up email. A member of our team will be in touch with a set of requests.

Best regards,

David M. Lewinsohn

Associate Editor

PLOS Pathogens

JoAnne Flynn

Section Editor

PLOS Pathogens

Kasturi Haldar

Editor-in-Chief

PLOS Pathogens

orcid.org/0000-0001-5065-158X

Michael Malim

Editor-in-Chief

PLOS Pathogens

orcid.org/0000-0002-7699-2064

Reviewer Comments (if any, and for reference):

Reviewer's Responses to Questions

**Part I - Summary**

Reviewer #1: The authors have adequately addressed my concerns.

Reviewer #2: The authors have done their best to address the issues we raised - and whilst they weren't able to modify their original patient cohort, they have gone some distance to increase the transparency of the data, specifically the baseline characteristics of the MDR/HIV cohort, and of those brought forward for further in-depth analysis within the MDR group.

This makes it much easier to put their findings into context.

The HIV+ group had - as we expected - significant viral load titres, and low CD4 counts. It's interesting to see that their findings in most of the data hold with the separation of HIV cohorts.

Rigor: Significant comment has been made about the nature of the observational study data here and how it fails to address mechanism. The analysis cannot identify important drivers, because the study is not designed to do so. The current draft of the abstract, at the moment, includes a comment in the penultimate sentence, which says

.... that metabolites and intermediates are "important drivers" of IL-1beta mediated proinflammatory signals.....

We suggest that the word "drivers" be removed, and use a word like "correlates", or "associations" please.

Reviewer #3: This is a revised submission that examines the linking TCA cycle remodeling with inflammation in humans infected with Mtb. The authors have sufficiently addressed my concerns either experimentally or with revised text. This manuscript helps further understand how host metabolism modulates Mtb disease and will be of great interest to the plos pathogen readership.

**Part II – Major Issues: Key Experiments Required for Acceptance**

Reviewer #1: (No Response)

Reviewer #2: (No Response)

Reviewer #3: (No Response)

**Part III – Minor Issues: Editorial and Data Presentation Modifications**

Reviewer #1: (No Response)

Reviewer #2: (No Response)

Reviewer #3: (No Response)

PLOS authors have the option to publish the peer review history of their article (what does this mean?). If published, this will include your full peer review and any attached files.

Reviewer #1: **Yes: **Maziar Divangahi

Reviewer #2: **Yes: **Joseph Keane

Reviewer #3: No

---

## [Editor Report · Acceptance letter]

21 Sep 2021

Dear Dr. Collins,

We are delighted to inform you that your manuscript, "TCA cycle remodeling drives proinflammatory signaling in humans with pulmonary tuberculosis," has been formally accepted for publication in PLOS Pathogens.

Best regards,

Kasturi Haldar

Editor-in-Chief

PLOS Pathogens

orcid.org/0000-0001-5065-158X

Michael Malim

Editor-in-Chief

PLOS Pathogens

orcid.org/0000-0002-7699-2064